# The von Willebrand Factor Antigen Reflects the Juvenile Dermatomyositis Disease Activity Score

**DOI:** 10.3390/biomedicines11020552

**Published:** 2023-02-14

**Authors:** Ellie Gibbs, Amer Khojah, Gabrielle Morgan, Louis Ehwerhemuepha, Lauren M. Pachman

**Affiliations:** 1Department of Biological Sciences, Wellesley College, Wellesley, MA 02481, USA; 2Department of Pediatrics, College of Medicine, Umm Al-Qura University, Makkah 21421, Saudi Arabia; 3Division of Pediatric Rheumatology, Ann & Robert H. Lurie Children’s Hospital of Chicago, Cure-JM Center of Excellence in Juvenile Myositis Research and Care, Chicago, IL 60611, USA; 4Computational Research, Children’s Hospital of Orange County Research Institute, Orange, CA 92868, USA; 5Feinberg School of Medicine, Northwestern University, Chicago, IL 60611, USA

**Keywords:** von Willebrand factor antigen, juvenile dermatomyositis, disease activity scores

## Abstract

Objective: This study determined if an accessible, serologic indicator of vascular disease activity, the von Willebrand factor antigen (vWF:Ag), was useful to assess disease activity in children with juvenile dermatomyositis (JDM), a rare disease, but the most common of the pediatric inflammatory myopathies. Methods: A total of 305 children, median age 10 years, 72.5% female, 76.5% white, with definite/probable JDM at diagnosis, were enrolled in the Ann & Robert H. Lurie Cure JM Juvenile Myositis Repository, a longitudinal database. Disease Activity Score (DAS) and vWF:Ag data were obtained at each visit. These data were analyzed using generalized estimating equation (GEE) models (both linear and logistic) to determine if vWF:Ag reflects disease severity in children with JDM. A secondary analysis was performed for untreated active JDM to exclude the effect of medications on vWF:Ag. Result: The vWF:Ag test was elevated in 25% of untreated JDM. We found that patients with elevated vWF:Ag had a 2.55-fold higher DAS total (CI_95_: 1.83–3.27, *p* < 0.001). Patients with difficulty swallowing had 2.57 higher odds of elevated vWF:Ag (CI_95:_ 1.5–4.38, *p* < 0.001); those with more generalized skin involvement had 2.58-fold higher odds of elevated vWF:Ag (CI_95_: 1.27–5.23, *p* = 0.006); and those with eyelid peripheral blood vessel dilation had 1.32-fold higher odds of elevated vWF:Ag (CI_95_: 1.01–1.72, *p* = 0.036). Untreated JDM with elevated vWF:Ag had more muscle weakness and higher muscle enzymes, neopterin and erythrocyte sedimentation rate compared to JDM patients with a normal vWF:Ag. Conclusion: vWF:Ag elevation is a widely accessible concomitant of active disease in 25% of JDM.

## 1. Introduction

Juvenile dermatomyositis (JDM) is a rare pediatric autoimmune disease with an incidence of approximately 3–4 cases/million children/year [1]. JDM is defined as a systemic vasculopathy [2]. It often presents with truncal and symmetrical proximal muscle weakness as well as a distinctive heliotrope rash that includes Gottron’s papules, occurring most often on the upper extremity, where the skin bends [3,4,5]. The diagnosis of JDM subtypes is facilitated by the relatively recent identification of protein-specific antibodies: the most common antibody in the Western world is anti-Tif-1-γ (p155/140), while anti-MDA-5 is dominant in the East [6,7,8]. JDM complications include macrophage activation syndrome [9]. Gastrointestinal damage may be as severe as pneumatosis intestinalis [10]. Other less common symptoms in JDM include weakened eye muscles, also specific targets in children with orbital myositis [11].

A variety of methods are available to assess JDM skin and muscle symptoms, including the disease activity score (DAS), used in this study, [12] or the childhood myositis assessment scale (CMAS) and the DASI [13]. More recently, the European League Against Rheumatism and American College of Rheumatology developed new classification criteria for JDM [14,15]. The DAS has 20 points and is divided into skin (0–9 points) and muscle (0–11 points) symptoms [12]. The skin evaluation includes an estimation of the extent of involvement, distribution of skin symptoms, vasculitis, and Gottron’s papules. Muscle weakness is usually associated with loss of specific muscle functional status as well [12,16]. In addition to the clinical tools, there is an urgent need for a robust, accessible, and inexpensive serological biomarker that indicates the severity of disease activity in children with JDM, especially with respect to vascular injury. The association of von Willebrand factor antigen (vWF:Ag) and the diagnosis of JDM had been previously reported [17,18]. However, the use of vWF:Ag as a reliable indicator of “disease activity” is not well established.

The aim of the present study was to examine the relationship between vWF:Ag and DAS components in children with JDM. We utilized a longitudinal data set collected over the whole disease course from the Cure JM Juvenile Dermatomyositis Registry, which includes data on over 500 children diagnosed with JDM. The first part of the study included an analysis of both treated and untreated JDM children, while the second part focused specifically on untreated JDM children to determine if an elevated vWF:Ag is associated with more severe JDM clinical manifestations than JDM with a normal vWF:Ag level.

## 2. Methods

### 2.1. Study Design, Approval, and Population

We created and utilized the Juvenile Dermatomyositis Registry (1971 through 2020) at Ann & Robert H. Lurie Children’s Hospital of Chicago, hereafter referred to as Lurie Children’s. Each patient in the registry has been diagnosed with definite/probable JDM, based on the Bohan and Peter criteria [4,19]. The data included in the registry of JDM were first obtained at the inception of the study in 1971, when we first recognized the extensive variability of clinical symptoms, and potential etiologies of JDM. There are over 1400 variables per child over time. Written patient informed consent was obtained for enrollment into the registry (Lurie Children’s Hospital IRB 2010-14117, last yearly approval 2 June 2022). Upon enrollment, data obtained throughout the course of routine clinical care for JDM is entered into a REDCap database. For this retrospective study, these secondary clinical data were queried for all patients meeting inclusion criteria, beginning in 1980 (through 2020), having at least six months’ follow-up data. Inclusion criteria were all subjects who had vWF:Ag level and definite/probable JDM according to the internationally accepted Bohan and Peter criteria (study was approved by the IRB of Lurie Children’s Hospital IRB 2008-1345, last yearly approval 28 August 2022). The Bohan and Peter diagnostic criteria include typical skin rash (heliotrope rash and Gottron’s sign) in addition to 3 out 4 of the following criteria: symmetric proximal muscle weakness, elevated serum muscle enzymes, abnormal electromyography and muscle biopsy [4,14,19]. We excluded subjects with overlap syndrome, such as positive anti-PM-Scl, anti-U1 RNP, or anti-U2 RNP, from the study. Of note, one patient was found to have a positive anti-Jo-1, 25 patients did not have myositis-specific or -associated antibody (MSA/MAA) data available, as their visits were prior to MSA/MAA testing being regularly tested around the year 2000, and one patient post-2000 did not have MSA/MAA data available due to a laboratory issue. Continually, 12 patients had one instance of indeterminate or possible U1RNP, U2RNP, U4RNP, U5RNP, U6RNP antibodies, with seven of these patients negative for the above antibodies upon retest.

### 2.2. Study Variables

The variables used in this study were vWF:Ag level, vWF:Ag category (normal, elevated), age, sex, race, duration of untreated disease, blood type, neopterin, erythrocyte sedimentation rate (ESR), muscle enzymes, T and B cell flow cytometry, treatment status at the first visit, and DAS components. As vWF:Ag is blood type-specific, the normal ranges are: type A = 48–234%, type B = 57–241%, type AB = 64–238%, and type O = 36–157%. The T and B cell flow cytometry was performed in the clinical immunology lab at Lurie Children’s at the time of diagnosis. Various antibodies manufactured by BD Biosciences (Franklin Lakes, NJ, USA) were used to evaluate CD45, CD3, CD4, CD8, CD16, CD56, and CD19. The DAS, assessed by a physician, is divided into skin (0–9 points) and muscle (0–11 points) components [12]. Both clinical assessment and laboratory testing were typically performed at the same visit. Generally, the cause of missing data was due to a failure by the clinician to collect the necessary information during patient visits or failure to order necessary laboratory tests for the patient. Patients meeting a threshold of less than 20% of missingness for any given variable were included. A secondary analysis was done for untreated active JDM (n = 138) to exclude the effect of medications on the vWF:Ag data (Figure 1).

### 2.3. Statistical Analysis

Data were analyzed using R software Version 4.1 in R Studio Version 1.4.1717. After variables were selected, summary statistics were obtained by finding the median or (count when appropriate) of demographic variables along with treatment status, DAS and vWF:Ag. Data that were consistent throughout a patient’s life (i.e., sex at birth and race) were evaluated for each patient, while data that changed depending on visit (such as vWF:Ag and DAS) were evaluated by visit. To assess the association of DAS subcomponents and vWF:Ag, vWF:Ag was stratified into “normal” and “elevated” for specific reference range for the subjects’ blood group. Two sets of models were built: (1) assessing DAS total, DAS skin, and DAS muscle as outcome variables, and (2) assessing the individual components of the DAS skin and DAS muscle scores. In the second model, vWF:Ag was selected as the outcome variable to simplify the analyses and preclude the need to build as many models as there are components of the DAS skin and muscle scores.

In other words, the association between DAS total, DAS skin, and DAS muscle, and vWF:Ag was tested using a generalized estimating equation linear model [20,21] that accounts for repeated measures (that is, the longitudinal nature of the data on disease activity). We used an unstructured correlation matrix [20,21] in this study to avoid making assumptions about the direction of correlations between repeated measures of the DAS. Final model selection was carried out using backward elimination controlling for demographics. Bonferroni correction was applied to adjust for multiple testing by multiplying each *p* value by corresponding number of tests across all models built.

A secondary analysis of the first visit data of untreated JDM patients (n = 138) was also performed to evaluate the utility of vWF:Ag, without the confounding effect of medications. In this analysis, we divided the study subjects into two groups (normal vWF:Ag and elevated vWF:Ag) based on the reference range of the vWF:Ag for the JDM child’s blood group. The Mann–Whitney test compared the medians of DAS, muscle enzymes, ESR, neopterin and flow cytometry results between the two groups. GraphPad Prism version 9.4.1 was used to generate the figures.

## 3. Results

### 3.1. Patient Background and Demographic Analysis

There were 393 JDM patients that met inclusion criteria for this study, but when assessed for missingness of less than 20% for any given variable, 305 patients with 5818 visits were included in the analysis. The average number of visits per patient was 19.1 (±14.0 SD), and median age for all visits of 10.7 (IQR = 7.3–14.9) years. The median age of registry enrollment was 7.1 (IQR = 4.4–10.3) years; 72.5% were female (n = 221), and most patients were white (76.5%), followed by Hispanic (15.7%), black (4.3%), Asian (2%), and other (1.3%) (Table 1). For those patients enrolled in the registry after initiation of treatment for JDM, many were referred from other centers and had variably reported medications prior to their first visit at Lurie Children’s; however, their mean duration of months from first therapy to registry enrollment was 9.0 (IQR = 1.3–29.9) (Table 1).

Since vWF:Ag levels differ depending on blood type, these data were collected and controlled for in later statistical tests. Most patients were blood type O (45%), followed by type A (36.7%), type B (13.1%), and AB (4.3%), frequencies that are relatively consistent with the general population [22]. The median total DAS for all visits was 4 out of 20 (IQR = 5.5), with the skin DAS median reported as 3 (IQR = 5) out of 9 and muscle DAS as 0 out of 11 (IQR = 2), and most patients had vWF:Ag levels available (89.9% in the overall database and 81% for the untreated JDM patients). It should be noted that age, DAS, and vWF:Ag medians were derived at the time of the encounter.

### 3.2. vWF:Ag Effect on DAS

We investigated the association between DAS total, DAS skin components, and DAS muscle components scores with demographic variables, duration of untreated disease, treatment status at first visit, and vWF:Ag to identify statistical associations. We found that age is associated with DAS total such that disease activity is highest at younger ages (effect size = −0.14, *p* < 0.001); having been treated elsewhere prior to their first visit at Lurie Children’s JM Clinic was associated with higher DAS total score with (effect size = 0.94, *p* = 0.014); and most notably, elevated vWF:Ag was associated with higher DAS total score (effect size = 2.55, *p* < 0.001) (Table 2). Differences in total DAS scores were not associated with sex (*p* = 0.624), duration of untreated disease (*p* = 1.000), or racial/ethnic group (*p* = 1.000).

For the DAS skin component scores, we found that higher scores were associated with African American race (effect size = −1.12, *p* = 0.009) than any other racial group (white as reference group, Hispanic *p* = 1.000, Asian *p* = 1.000, other race *p* = 0.677) and higher vWF:Ag levels (effect size = 0.96, *p* < 0.001) (Table 1). Neither duration of untreated disease (*p* = 0.726) nor sex (*p* = 0.485) was associated with DAS skin.

Finally, for muscle DAS components, we found that age is associated with DAS muscles such that it is highest at younger ages (effect size = −0.13, *p* < 0.001). Having been treated at the first visit was associated with higher DAS skin scores (effect size = 0.52, *p* < 0.005), and elevated vWF:Ag was also associated with higher DAS muscle scores (effect size = 1.59, *p* < 0.001) (Table 2). Differences in DAS muscle scores were not associated with sex (*p* = 1.000), race (white as reference group: black/African American, *p* = 0.084; Hispanic, *p* = 1.000; Asian, *p* = 1.000; other race, *p* = 1.000), and duration of untreated disease (*p* = 1.000).

For all components of DAS, no significant relationship with blood type was found (type O as reference group: DAS total and muscle, all blood types, *p* = 1.000; DAS skin, type A, *p* = 0.979; type B, *p* = 1.000; type AB, *p* = 1.000). When comparing the effect size of muscle and skin components, we find that an elevated vWF:Ag has a higher association with muscle DAS components (effect size = 1.59, *p* < 0.001) than with skin symptoms (effect size = 0.96, *p* < 0.001) as shown in Table 2.

### 3.3. Subcomponents Associated with vWF:Ag

We subsequently assessed subcomponents of the DAS to determine their association with the increased vWF:Ag levels. We found that skin involvement at specific focal points (i.e., a joint-related skin rash) had a 46% increased odds of higher vWF:Ag levels (*p* = 0.013), generalized skin involvement had 158% increased odds (*p* = 0.006), and eyelid peripheral blood vessel dilation had 32% increased odds (*p* = 0.036) (Table 3).

Data concerning muscle weaknesses, such as the inability to clear scapula (56% increased odds, *p* < 0.001), lower proximal muscle weakness (40% increased odds, *p* = 0.009), Gower’s sign (58% increased odds, *p* = 0.004), and difficulty swallowing (157% increased odds, *p* < 0.001), were all associated with a higher vWF:Ag (Table 3). Muscle functional status in children with JDM, especially with respect to the impact on their lives, was also significantly associated with vWF:Ag data at all levels. JDM with minimal limitations had 81% increased odds of higher vWF:Ag (*p* < 0.001), those with mild limitations had 159% increased odds of higher vWF:ag levels (*p* < 0.001), while JDM with moderate limitations had 265% increased odds (*p* < 0.001) and JDM with severe limitations had a 819% increased odds (*p* < 0.001). Therefore, there is a strong correlation between increasing levels of the severity of the loss of muscle function and increased odds of a higher value for vWF:Ag in a given individual.

Of note, age did not significantly increase odds of higher vWF:Ag (*p* = 1.000), nor male sex (*p* = 1.000) or race/ethnic group (white as reference group; black/African American, *p* = 0.156; Hispanic, *p* = 1.000; Asian, *p* = 1.000; other race, *p* = 0.170). Additionally, while vWF:Ag reference range is blood type-specific, no blood type had significantly increased odds of higher vWF:Ag (type O as reference group: type A, *p* = 0.053; type B, *p* = 0.149; type AB, *p* = 0.361).

### 3.4. Untreated JDM with Active Disease

Many patients arrive at Lurie Children’s before receiving any medical treatment. In this study, 45.2% (n = 138) were untreated at registry enrollment, their median duration of untreated disease was 4.3 months (IQR = 2.0–10.3), median age at diagnosis was 6.1 (IQR = 4.0–9.4) years and 76.1% (n = 105) were female (Table 1). For untreated active JDM, 25% (35 of 138) had elevated vWF:Ag levels. Patients with increased vWF:Ag levels were older and had shorter duration of untreated disease than those with normal vWF:Ag levels (Table 4). As expected from the previous analysis, the groups with elevated vWF:Ag than normal vWF:Ag had higher median total DAS (13 vs. 11, *p* < 0.0001) and muscle DAS (8 vs. 5, *p* < 0.0001) (Figure 2). In addition, muscle enzymes such as creatine phosphokinase, aspartate aminotransferase, lactate dehydrogenase and aldolase (Figure 3) were higher in the elevated vWF:Ag group (Table 5). The inflammatory markers neopterin and ESR were higher in the elevated vWF:Ag group as well (Figure 4A,B,C and Table 5). Lymphocyte subset analyses by flow cytometry showed that an elevated vWF:Ag was associated with a decreased NK (natural killer) cell count than normal vWF:Ag (73 vs. 140 cells/mm^3^, *p* = 0.0005) (Figure 4C).

## 4. Discussion

In this study, we document a significant correlation between vWF:Ag level and disease activity score in JDM. Interestingly, muscle DAS components had a greater association with elevated vWF:Ag levels than skin DAS, especially in the untreated subjects. These data have also shown that untreated children with elevated vWF:Ag had more muscle weakness and higher muscle enzymes than JDM patients with normal vWF:Ag, documenting the utility of this biomarker in revealing muscle involvement. Furthermore, both neopterin and erythrocyte sedimentation rate were more elevated in the high vWF:Ag group, which confirms the association of vWF:Ag with other indicators of inflammation in children with JDM. This study’s results align with findings from a previous study that demonstrated that vWF:Ag levels are higher in JDM patients with increased serum neopterin, a marker of macrophage activation [23]. vWF:Ag has also been associated with downregulated miRNA-10a, a proinflammatory mediator [24]. RNA transcriptomes have shown that JDM can be associated with evidence of increased inflammation, even when the child, on examination, appears to be clinically inactive [25]. The appearance of clinical quiescence usually triggers reduction in immunosuppressive medication, resulting in cycles of disease flares and prolonging disease chronicity. Therefore, utilizing vWF:Ag level and other inflammatory markers in clinical practice may help in recognizing disease activity in the apparently clinically quiescent patients.

In this study, JDM patients with eyelid marginal blood vessel dilation were 32% more likely to have elevated vWF:Ag (Table 3), supporting the use of vWF:Ag as a biomarker for vasculitis in JDM. This finding is consistent with previous research linking elevated vWF:Ag to active CNS vasculitis and ANCA vasculitis [26,27,28], which is not unexpected as vWF:Ag is primarily synthesized in the endothelial cells and megakaryocytes [27,28,29]. Additionally, elevated vWF:Ag activity has been linked to a greater likelihood of severe COVID-19 requiring ICU admissions [30] and it also predicts portal hypertension and mortality in patients with liver cirrhosis [31,32]. We did not find any significant difference in the mean vWF:Ag level for JDM patients diagnosed before and after 1 January 2020, which corresponds to the date of the initial cases of COVID-19 in Chicago (manuscript under review).

Although vWF:Ag is a relatively inexpensive indicator of endothelial cell damage [33,34], it is not always reliable for the diagnosis of JDM, as a majority of untreated JDM patients still have normal vWF:Ag levels. Nailfold capillary loss, a common feature of JDM, is a more reliable indicator of small vessel injury [3,35]. Notably, baseline vWF:Ag varies depending on blood group, with group O having the lowest baseline and group AB having the highest [36]. Other current biomarkers for JDM, such as muscle enzymes and inflammatory markers, also have limitations [37]. For example, although elevated muscle enzymes are an excellent tool early in the disease course, they tend to normalize in patients with untreated disease for ≥4.7 months [38]. Traditional inflammatory markers such as ESR and C-reactive protein are not useful for most children with JDM, as most patients have a normal ESR [23]. On the other hand, neopterin, a small molecule produced by activated macrophages, is a good marker of disease activity, but has limited availability in commercial labs [23]. In summary, to accurately assess JDM disease activity, a thorough physical examination and the use of multiple biomarkers should be utilized to address the limitations of any single marker.

Study limitations include the following. Only 25% of untreated JDM had an elevated vWF:Ag. However, those patients who did have an elevated vWF:Ag had more severe symptoms, such as muscle weakness and difficulty swallowing, as well as higher levels of inflammatory markers. A significant portion of the primary cohort (55%, n = 305) had been treated at other centers, and the details of their treatment and initial clinical presentation were unclear. Therefore, we conducted a secondary analysis on the untreated group (n = 138) to avoid confounding effects of medication use.

## 5. Conclusions

These results establish vWF:Ag as an accessible but not very sensitive biomarker for JDM diagnosis. Potentially, this assay can be employed to monitor disease activity in order to avoid premature termination of medical therapy. Finally, vWF:Ag is an affordable and readily available test, making it a possible biomarker for clinicians to consider, even in geographic areas with limited availability of resources.

## Figures and Tables

**Figure 1 biomedicines-11-00552-f001:**
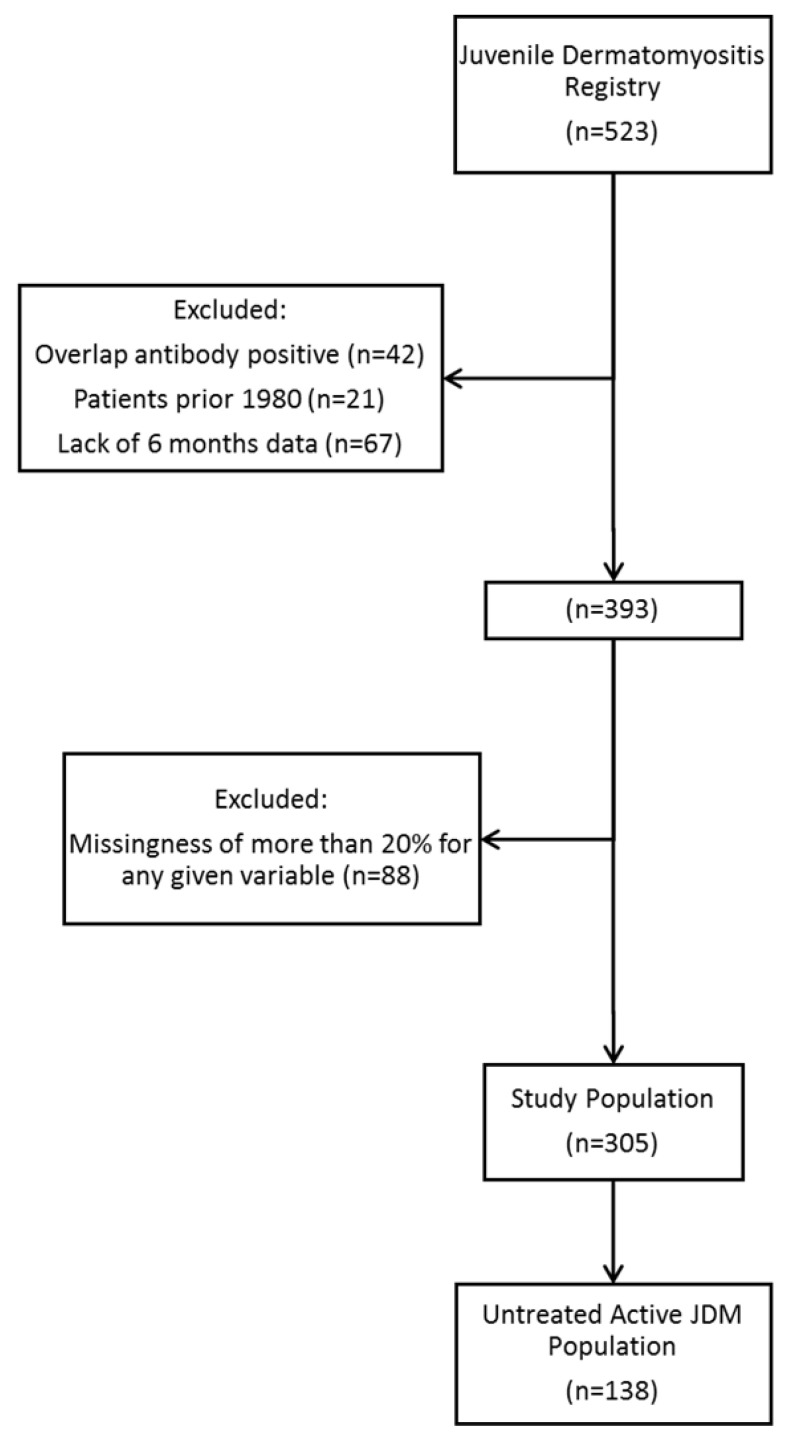
Inclusion criteria flow diagram. Patients selected from the Juvenile Dermatomyositis Registry at Lurie Children’s meeting inclusion criteria for the study.

**Figure 2 biomedicines-11-00552-f002:**
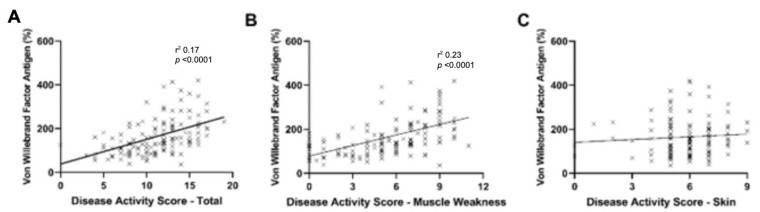
Untreated JDM vWF:Ag levels and DAS. Untreated JDM patients with higher vWF:Ag had higher (**A**) DAS total and (**B**) DAS muscle weakness, but not (**C**) DAS skin.

**Figure 3 biomedicines-11-00552-f003:**
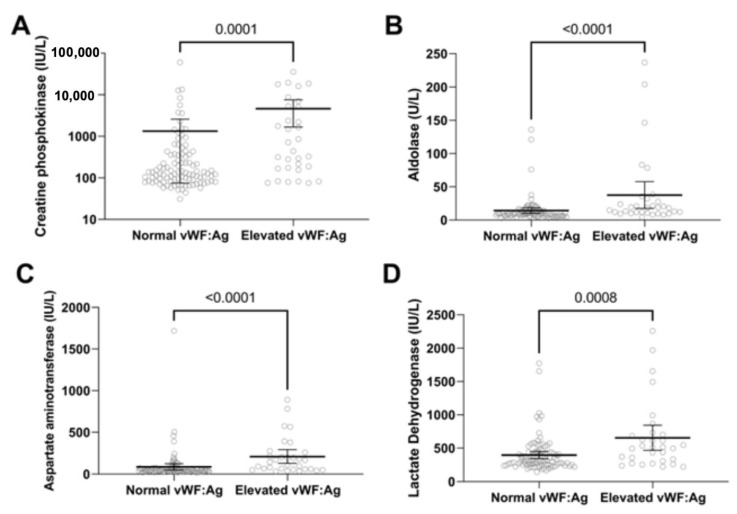
vWF:Ag groups and muscle enzyme levels in untreated JDM. JDM disease activity indicators in untreated JDM (n = 138) with normal vWF:Ag and elevated vWF:Ag. Untreated JDM with elevated vWF:Ag showed (**A**) higher creatine phosphokinase levels, (**B**) higher aldolase levels, (**C**) higher aspartate aminotransferase levels, and (**D**) higher lactate dehydrogenase levels.

**Figure 4 biomedicines-11-00552-f004:**
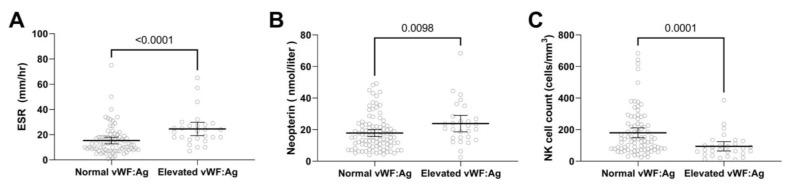
vWF:Ag groups and inflammatory markers in untreated JDM. JDM disease activity indicators in untreated JDM (n = 138) with normal vWF:Ag and elevated vWF:Ag. Untreated JDM with elevated vWF:Ag showed (**A**) higher ESR levels, (**B**) higher neopterin levels, and (**C**) lower NK cell absolute counts.

**Table 1 biomedicines-11-00552-t001:** Demographics and Clinical Characteristics of 305 Patients with JDM.

	Total JDM Population n = 305	Untreated JDM n = 138
Demographics		
Age at enrollment (yr), median (IQR)	7.1 (4.4–10.3)	6.1 (4.0–9.4)
Age at onset of symptoms (yr), median (IQR)	5.6 (3.1–8.4)	5.5 (3.4–8.4)
Sex		
Male, n (%)	84 (27.5)	33 (23.9)
Female, n (%)	221 (72.5)	105 (76.1)
Race/Ethnicity		
White, n (%)	234 (76.7)	101 (73.2)
Hispanic, n (%)	48 (15.7)	25 (18.1)
Black/African American, n (%)	13 (4.3)	6 (4.3)
Asian, n (%)	6 (2.0)	5 (3.6)
Other, n (%)	4 (1.3)	1 (0.7)
Clinical Factors	median (IQR)	median (IQR)
Duration of untreated disease (mo)	4.3 (2.0–10.3) n = 302	6.0 (3.1–12.8)
Duration from 1st treatment to enrollment ^a^ (mo)	9.0 (1.3–29.9)	n/a
DAS		
DAS Total (0–20)	11.0 (7.5–13.0) n = 273	12.0 (8.9–14.0) n = 133
DAS Skin (0–9)	6.0 (5.0–7.0) n = 273	6.0 (5.0–7.0) n = 133
DAS Muscle (0–11)	5.0 (2.0–8.0) n = 276	6.0 (3.0–8.0) n = 133
Laboratory Disease Activity Indicators		
Neopterin (nmol/liter)	12.9 (7.0–19.4) n = 257	17.2 (11.0–23.9) n = 124
ESR (mm/hr)	12.0 (7.0–20.0) n = 218	15.0 (9.0–23.0) n = 104
vWF:Ag (%)	140.0 (99.8–203.5) n = 260	140.0 (99.0–206.3)
Muscle Enzymes		
CK (IU/L)	114.0 (71.0–341.5) n = 267	154.5 (85.5–631.8) n = 132
LDH (IU/L)	298.5 (223.0–433.3) n = 264	350.5 (264.0–527.3) n = 131
AST(SGOT) (IU/L)	40.5 (28.0–67.3) n = 250	49.0 (36.0–107.0) n = 125
Aldolase (U/L)	8.8 (6.3–14.0) n = 249	10.3 (7.6–18.0) n = 123

^a^ Patients enrolled after starting treatment for JDM.

**Table 2 biomedicines-11-00552-t002:** Association of DAS components (total, skin, and muscle) with JDM demographic and disease factors ^a^.

	DAS Total		DAS Skin		DAS Muscle	
	Effect Size (95% CI)	*p* Value	Effect Size (95% CI)	*p* Value	Effect Size (95% CI)	*p* Value
Age (yr)	−0.14 (−0.21, −0.06)	<0.001	−0.01 (−0.06, 0.04)	1.000	−0.13 (−0.16, −0.09)	<0.001
Treatment Status at 1st Visit						
Untreated	Reference		Reference		Reference	
Treated	0.94 (0.13, 1.76)	0.014	0.42 (−0.18, 1.02)	0.385	0.52 (0.11, 0.94)	0.005
vWF:Ag						
Elevated	2.55 (1.83, 3.27)	<0.001	0.96 (0.59, 1.34)	<0.001	1.59 (1.13, 2.04)	<0.001

^a^ All reported statistics adjusted for multiple comparison using Bonferroni correction.

**Table 3 biomedicines-11-00552-t003:** Association of specific DAS components with vWF:Ag levels using odds ratios.

	Odds Ratio (95% CI)	*p* Value
Duration of untreated disease (mo)	0.95 (0.91, 0.995)	0.025
Treatment status at 1st visit		
Untreated	Reference	
Treated	1.03 (0.66, 1.61)	1.000
DAS Skin Elements		
Skin Involvement Distribution		
None	Reference	
Focal (including areas of joint related skin)	1.46 (1.07, 2.00)	0.013
Diffuse (including extensor surfaces of limbs/shawl)	1.38 (0.88, 2.17)	0.211
Generalized (including trunk involvement)	2.58 (1.27, 5.23)	0.006
Eyelid blood vessel dilation, present	1.32 (1.01, 1.72)	0.036
DAS Muscle Elements		
Functional Status		
Within Normal Limits	Reference	
Minimal Limitations	1.81 (1.27, 2.58)	<0.001
Mild Limitations	2.59 (1.81, 3.70)	<0.001
Moderate Limitations	3.65 (2.14, 6.23)	<0.001
Severe Limitations	9.19 (3.33, 25.38)	<0.001
Neck flexor weakness	1.32 (0.997, 1.75)	0.053
Unable to clear scapula	1.56 (1.22, 2.01)	<0.001
Lower proximal muscle weakness	1.40 (1.07, 1.83)	0.009
Gower’s Sign	1.58 (1.13, 2.19)	0.004
Difficulty Swallowing	2.57 (1.50, 4.38)	<0.001

**Table 4 biomedicines-11-00552-t004:** vWF:Ag in 138 untreated children with JDM: demographic characteristics.

	Normal vWF:Ag	Elevated vWF:Ag	*p* Value
Number of subjects, n (%)	103 (74.6%)	35 (25.4%)	
Age at onset of symptoms in years, median (IQR)	5.7 (3.9–8.7)	8.4 (5.4–9.9)	0.017
Duration of untreated disease in months, median (IQR)	6.4 (3.6–13.5)	3.8 (1.9–7.2)	0.004
Sex, n (%)			
Female	76 (74%)	29 (83%)	0.277
Male	27 (26%)	6 (17%)	
Race/ethnicity, n (%)			
White	79 (77%)	22 (63%)	0.184
Hispanic	17 (16%)	3 (7%)	
African American	3 (3%)	8 (23%)	
Asian	4 (4%)	1 (3%)	
Others	0 (0%)	1(3%)	

**Table 5 biomedicines-11-00552-t005:** vWF:Ag levels in 138 untreated children with JDM: disease activity markers.

	Normal vWF:Ag	Elevated vWF:Ag	*p* Value
	Median (IQR)	Median (IQR)	
DAS			
DAS Total	11 (8–13)	13 (12–16)	<0.0001
DAS Skin	6 (5–7)	5.5 (5–7)	0.704
DAS Muscle	5 (2.5–7)	8 (6–9)	<0.0001
Laboratory Disease Activity Indicators			
Neopterin (nmol/L)	15.3 (9.5–22.4)	21.6 (14.8–28)	0.0098
ESR (mm/h)	13 (8–18)	23	<0.0001
Muscle Enzymes			
CK (IU/L)	130 (83–356)	579 (169–5283)	0.0001
LDH (IU/L)	315 (253–435)	512 (307–703)	0.0008
AST(SGOT) (IU/L)	44 (33–64)	143 (58–258)	<0.0001
Aldolase (U/L)	9.6 (6.9–14.2)	18.4 (11–31)	<0.0001
Flow Cytometry (% lymphocytes)			
Total T cells (CD3+)	62.9 (57–68)	69 (63–74)	0.0008
T helper cells (CD3+ CD4+)	42 (37–48)	48 (43–54)	0.0006
T cytotoxic cells (CD3+ CD8+)	19 (16–22)	20 (17.5–23)	0.323
B cells (CD19+)	29.5 (24–35)	25 (19.8–30.8)	0.049
NK cells (CD16+/CD56+)	6 (4–9)	4 (3–7)	0.0097

## Data Availability

Data for study is not publicly available. Please reach out to corresponding author for further inquiries.

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
