# Peer review of "The von Willebrand Factor Antigen Reflects the Juvenile Dermatomyositis Disease Activity Score"

_biomedicines, 2023, doi:10.3390/biomedicines11020552_

Round 1
Reviewer 1 Report
INTRODUCTION
- In the first paragraph, in addition to the general clinical aspects of JDM, the authors should mention some details about the potential inflammatory complications of JDM, including the possibility of MAS onset (refer to: Macrophage activation syndrome in juvenile dermatomyositis: a systematic review. Rheumatol Int. 2020 May;40(5):695-702. doi: 10.1007/s00296-019-04442-1.), as it may happen in the other main pediatric rheumatic disorders.
- In the first paragraph, the authors supported 3 sentences with the same references n.5: [5] can be placed just once at the end of all these three sentences. In this regard, ref. [5] is quite dated (2003) and, since then, JDM diagnostic criteria and also immunological aspects (including autoantibodies) have been further clarified and should be mentioned, too. Therefore, more updated references, also describing these aspect, should be used (e.g. Clinical and autoantibody phenotypes of juvenile dermatomyositis. Reumatologia. 2022;60(4):281-291. doi: 10.5114/reum.2022.119045)
- The second paragraph (“There is a need…clinically quiescent”) could be summarized and provide a general background only. Some details described here may better fit the discussion later, perhaps.
- The final sentence, including the general study objectives, should represent a separate paragraph from the previous one. Moreover, I would suggest the authors expand a little the definition of the study objectives at this point.
METHODS
- I would suggest naming the first subsection as “study design and population” and, then, complete it accordingly.
- I also suggest creating a specific subsection for the ethical statements. Concomitantly, I would like the authors to clarify the informed consent processing (“Written patient informed consent was obtained”). Indeed, this study seems to be mainly a retrospective studies with the first patients enrolled in this registry in 1971, if my understanding is correct. In this regard, it is really important that the authors better describe the study design, as commented above.
- “The variables used in this study were: vWF:Ag level, vWF:Ag category (normal, elevated), age, sex, race, duration of untreated disease, blood type, neopterin, Erythrocyte Sedimentation Rate (ESR), Muscle enzymes, T and B cell flow cytometry, treatment status at the first visit, and DAS components.”: data collection strategy should be clarified. I assume these are all secondary existing data, is it correct? Anyway, the technical specifications of analyses that are not made on a routine basis should be defined, especially as regards the core analyses related to the vWF. Moreover, T and B cell flow cytometry should be clearly described. Finally, it is important to clarify if any analyses were performed specifically for this research purposes or all were secondary data, as mentioned above.
- The authors should also clarify the blood sample timing, as regards the laboratory parameters included in their analysis.
- Inclusion and exclusion study criteria should be clearly defined. The authors should also clarify the diagnostic criteria used for JDM diagnosis.
- Overall, this section should be re-organized and completed, according to the previous comments.
RESULTS
- I think the authors should add a first table showing the detailed demographic and clinical features (including DAS, general laboratory parameters and JDM main manifestations, especially according to the diagnostic criteria) in all the study population.
- In the first sentence, state the total number of JDM patients included in this study. All the numerical demographic parameters (e.g. age) should be expressed with a decimal and the dispersion should be defined.
- In table 1, the authors reported that some patients were treated and others untreated at the first visit and, I guess, at the study enrollment. Can you clarify this point and, probably, this aspect should be considered in the new and additional table that I suggested above. If so, some details about therapeutic regimen and duration before enrollment should be provided.
- Probably, Table 3 could be revised and enriched according to the additional information requested for the new initial table.
- I suggest the authors retrieve and add the ferritin values as well, as an important chronic inflammatory parameter.
DISCUSSION
- I recommend the authors start the discussion by listing, summarizing, and emphasizing the main findings emerging from their results.
- These aspects, if several, should be analyzed one by one in light of the existing literature.
- The discussion can be fully reviewed after the authors address the methodological and results comments and, perhaps, start rearranging the discussion accordingly.
- At some point, the authors mention COVID-19: by the way, I also suggest the authors to clarify in the methods the exact study period, especially as regards the ending month in 2020.
- In the current version, the discussion sounds quite dispersive, but the authors could improve it by ordinately discussing the main findings as suggested above.
- I think the limitations are more than those currently reported by the authors. Indeed, some methodological aspects to be clarified may represent additional limitation.
- The authors should create a specific and clear conclusion section.
- As mentioned, the review of the discussion cannot be completed, since methodological clarification and results completions are definitely needed.
Reviewer 2 Report
Dear Authors,
Well written and understandable manuscript about the important topic. I have just some minor remarks for you:
1) please, avoid Grammar and careless errors (spaces, redundant punctuation), they do not fit for this manuscript;
2) please, add the Conclusions at the end of manuscript;
3) you have not very many Literature sources - only 28. So, are you sure that you have to include these 3 previous century references!? Could you remove/change them with an other ones - more modern ones, please.
Otherwise thank you for this manuscript, it was interesting to read it!
Reviewer 3 Report
General Comments:
Thank you for submitting an interesting paper. I have some questions and comments.
Major points:
1. This study covered only definite cases, and excludes probable and possible cases, according to the Bohan and Peter criteria?
2. The description of the discussion is insufficient, please make it more complete.
3. Please emphasize the novelty of the results of this study.
Minor comments:
4. The von and the Von are mixed, so please use one of them.
5. The vWF:ag and the vWF Ag are mixed, so please use one of them.
6. All abbreviations and acronyms should be expanded, followed by the abbreviation or acronym in parentheses, upon first use in the abstract, as well as in the first use in the body of the manuscript. All subsequent uses, including tables and figures, should use the abbreviation or acronym. For example, please review Juvenile Dermatomyositis/JDM.
7. Why don't you move the discussion from VWD to liver cirrhosis to the introduction?
8. Please add the reference to the Bohan and Peter criteria.
9. Please add the normal range of vWF:Ag in the method.
10. Why are B cells and NK cells decreased in the elevated vWF:Ag group?
Round 2
Reviewer 1 Report
The authors provided detailed explanations and comments to all my points. Therefore, I have no additional major revisions to suggest.
Author Response
We thank the reviewer for their reviews of our manuscript.
Reviewer 3 Report
1. Please add a Flow diagram of the subject selection process as a figure.
2. Were all 393 JDM patients tested for anti-PM-Scl, anti-U1 RNP, or anti-U2 RNP?
3. Please add the location of BD Biosciences.
4. They are abbreviated as CK, AST, and LDH in Table 1, but are not abbreviated as Creatine phosphokinase, Aspartate aminotransferase, and Lactate Dehydrogenase in Table 5. Please unify. Also, the order is CK, LDH, and AST in Table 1, but it is CK, AST, and LDH in Table 5. Please unify. Moreover, in Table 1, the order is Muscle enzymes, then Laboratory data, whereas in Table 5, the order is Laboratory data, then Muscle enzymes. Please unify.
5. In Table 5, (CD3+ CD4 should be (CD3+ CD4+).
6. Consistently use uppercase and lowercase letters throughout the manuscript. Examples include Muscle Enzymes and Muscle enzymes, vWF:Ag in 138 Untreated Children with JDM: Demographic Characteristics and vWF:Ag levels in 138 untreated children with JDM: Disease activity markers.
7. Please unify the writing of Age at enrollment (yr), median (IQR) and Age at onset of symptoms in years, median (IQR).
8. Part of Discussion is in bold, thus please fix it.
